# Loneliness and Life Satisfaction Explained by Public-Space Use and Mobility Patterns

**DOI:** 10.3390/ijerph16214282

**Published:** 2019-11-04

**Authors:** Lisanne Bergefurt, Astrid Kemperman, Pauline van den Berg, Aloys Borgers, Peter van der Waerden, Gert Oosterhuis, Marco Hommel

**Affiliations:** 1Department of the Built Environment, Eindhoven University of Technology, 5600MB Eindhoven, The Netherlands; a.d.a.m.kemperman@tue.nl (A.K.); p.e.w.v.d.berg@tue.nl (P.v.d.B.); a.w.j.borgers@tue.nl (A.B.); p.j.h.j.v.d.waerden@tue.nl (P.v.d.W.); 2Adviesbureau PLANTERRA, 3833GL Leusden, The Netherlands; gert.oosterhuis@planterra.nl (G.O.); marco.hommel@planterra.nl (M.H.)

**Keywords:** public space, neighborhood, loneliness, life satisfaction, mobility, elderly, path analysis

## Abstract

Previous research has shown that personal, neighborhood, and mobility characteristics could influence life satisfaction and loneliness of people and that exposure to public spaces, such as green spaces, may also affect the extent to which people feel lonely or satisfied with life. However, previous studies mainly focused on one of these effects, resulting in a lack of knowledge about the simultaneous effects of these characteristics on loneliness and life satisfaction. This study therefore aims to gain insights into how public-space use mediates the relations between personal, neighborhood, and mobility characteristics on the one hand and loneliness and life satisfaction on the other hand. Relationships were analyzed using a path analysis approach, based on a sample of 200 residents of three neighborhoods of the Dutch city ‘s-Hertogenbosch. The results showed that the influence of frequency of public-space use on life satisfaction and loneliness is limited. The effects of personal, neighborhood, and mobility characteristics on frequency of use of public space and on loneliness and life satisfaction were found to be significant. Age and activities of daily living (ADL) are significantly related to each other, and ADL was found to influence recreational and passive space use and loneliness and life satisfaction. Policymakers should, therefore, mainly focus on creating neighborhoods that are highly walkable and accessible, where green spaces and public-transport facilities are present, to promote physical activity among all residents.

## 1. Introduction

The belief that exposure to nature, such as trees and water, promotes well-being and life satisfaction, dates as far back as the rise of the first cities. The ancient residents of Rome wrote, for instance, that they valued the contact with nature to escape from the noise and the congestion of the city [1]. In the urbanizing society in which we live nowadays, there is a growing need for nature and public open spaces to escape from our hectic city lives [2]. Within thirty years from now, almost 70% of the world’s population is expected to live in urban areas, increasing the demand of high-quality public spaces [3]. In recent years, however, public spaces seem to have diminished or reduced in quality, specifically in highly urbanized areas [4].

The increase in population density in urban areas results in the poorer health and well-being of its residents [5]. People who live in green environments, generally, are more positive about their self-perceived health [4] and about their life satisfaction. Furthermore, people who live in green environments feel less lonely than people who live further away from green spaces [6]. The effect of green spaces on life satisfaction and loneliness is even stronger for people who are assumed to spend more time at their homes, such as elderly, children, and people with a low economic status [4]. Here, life satisfaction can be defined as a person’s cognitive overall assessment of his or her life [7]. Subsequently, loneliness is defined as the discrepancy between an individual’s achieved and desired level of social relationships [8].

Public spaces, such as parks, squares, streets, play areas, and civic spaces, can promote social interactions [9]. Social interactions and community involvement are related to life satisfaction. Residents who invest time in their community tend to be happier and to report their life satisfaction to be higher [10]. For elderly people, specifically, having opportunities for social contact in public spaces positively affects life satisfaction [11]. For ageing in places to work well, neighborhoods should facilitate amenities, such as recreational and sports facilities and green spaces, where social interactions between neighbors are promoted [12,13,14,15].

With an expected increase in the number of elderly people, from 0.9 million people above 65 in 2018 to 1.8 million in 2060 in the Netherlands [16], renewed interest in the effect of quality of public spaces on people’s life satisfaction and loneliness arises [17]. Both the increase in the number of elderly and the urbanization in future decades ask for a strategy that promotes life satisfaction and decreases loneliness of citizens. Since ageing residents have a smaller action radius [13], it is imperative to understand which characteristics of public space can explain the use of space, increase life satisfaction, and reduce loneliness. This requires an analysis in which personal, mobility, and social neighborhood characteristics are used simultaneously to understand loneliness and life satisfaction of residents. This study therefore aims to gain insights into which personal, mobility, and social neighborhood characteristics influence the use of public space, loneliness, and life satisfaction of residents.

## 2. Literature Review

In recent years, people have started to live more isolated from others, live to an older age, have fewer children, divorce more often, and live further away from friends and family for education and careers. These developments all contribute to people feeling socially excluded and, consequently, lonely [18]. There is growing recognition that not only personal characteristics, such as being older or being healthy, but also neighborhood characteristics, affect the loneliness of people. The subjective feelings about a neighborhood can also be a significant source of life satisfaction [19,20] and loneliness [21].

In addition to the personal and social neighborhood characteristics, mobility characteristics, such as transport-mode use, frequency of visits, and distance from public spaces, were found to affect loneliness [22] and life satisfaction [6,23]. For elderly people, in particular, being able to walk and cycle in public spaces, to meet family and friends outdoors [24], and to have social interactions reduces the risk of becoming lonely [21]. This section continues to discuss the personal and social neighborhood characteristics and the mobility patterns that were found to influence public-space use, life satisfaction, and loneliness.

### 2.1. Personal Characteristics

Personal characteristics such as age, gender, ethnic background, income, educational background, and household composition are often considered to affect the life satisfaction and loneliness of people. Age was found to be related to life satisfaction, mediated by the effect of health. People who are less healthy reported lower life satisfaction than healthy people [25]. Ageing people, specifically, are more likely to report lower health and are therefore also more likely to be unsatisfied with life [26]. The relationship with loneliness also indicates that the elderly are the most lonely and that loneliness increases over time [27]. The differences in life satisfaction between males and females is found by some authors, who argue that women are generally more satisfied with life than men [28,29]. Women are also found to feel lonelier than man, especially after the age of 55. Previous research has shown that the increase in loneliness over time is triggered by different life events and experiences, such as losing a partner [30]. It was, moreover, found that differences in life satisfaction exist between people with different ethnic backgrounds, and among males as well as females [29].

Next, the educational background and income of residents are often discussed as factors that influence life satisfaction and loneliness. Among the youngest generations, investing in education is seen as a method to generate wealth and become happier [31]. For the elderly, the relationship between education and satisfaction was not found. Elderly people tend to have lower educational backgrounds but tend to be more satisfied with their life [32]. Some authors therefore argue that income is a stronger contributor to life satisfaction than education. Generally, higher income leads to higher life satisfaction [23,33]. Having a higher income enables people to use paid services, resulting in improved life conditions and human well-being and higher life satisfaction [34]. People who can afford paid services have more opportunities of social interaction and participation and are at lower risk of becoming lonely [35,36].

Having a higher level of education or a larger income reduces the risk of becoming lonely. Other protective factors, such as having a larger household size, being married, and being in good health, limits the vulnerability of people to feeling lonely [30]. For elderly people in particular, the household composition is a strong predictor of loneliness. Being married or living in larger households, instead of living alone, protects them from feeling lonely [30]. Elderly people who live with a partner are more satisfied with life than people who live alone [37]. People who live alone are more likely to be dependent on others, particularly when they are in poor health [38]. Among the oldest residents of a neighborhood, being able to do daily activities independently, such as bathing, dressing, and eating, contributes to life satisfaction [37]. 

### 2.2. Mobility

Mobility characteristics consist of the travel behavior and activeness of people. People who make more daily out-of-home trips are, in general, more satisfied with their life [10]. People who travel to places with active modes of transport or with the car are more satisfied with their daily travels than people who use public transport [39]. Residents who make daily trips with different transport modes are also observed to be more satisfied with their life [10]. Having access to different transport modes, such as public transport and a car, increases the opportunities to participate and to interact with others. People who own a car or who use a car are also found to be less lonely than people with limited access to mobility [21].

Another characteristic of mobility, which extends beyond the type of mobility or means of travel and does also include a type of sociability between neighbors, is walkability [40]. People who live in walkable neighborhoods participate more often in events, know their neighbors, and are socially engaged. The extent to which these people perceive themselves as community members mediates the effect between walkability and life satisfaction [41]. Residents of walkable, green neighborhoods have more social interactions [42] and are less prone to becoming lonely than people who live in less-green environments [22].

### 2.3. Social Neighborhood Characteristics

Social neighborhood indicators are, amongst others, neighborhood attachment and social cohesion. For the ageing population, frequent contact with neighbors contributes to their social network [43]. Elderly people who participate in social activities in a neighborhood are more satisfied with their life than elderly people who participate in less-social activities [44]. Feeling attached to the neighborhood invites people to actively participate in the public space and to have social interactions. People who are attached to the neighborhood are more satisfied with their life [6] and feel less lonely than people who do not feel attached [45]. 

The contacts and emotional connections between neighbors, measured as social cohesion, contribute to life satisfaction [46]. Especially among vulnerable groups of people, social cohesion in a neighborhood is a significant indicator of life satisfaction [47]. People who live in neighborhoods with high levels of social cohesion report higher numbers of social interactions [45]. Public spaces where people can meet and get acquainted with the neighborhood stimulate social cohesion [48]. A lack of recreational facilities, sports facilities, and green spaces may result in less social interactions [13], resulting in weaker reported social cohesion [49] and a greater risk of vulnerable residents becoming lonely [50]. For elderly people, being able to access facilities and services enables them to stay independent [51].

## 3. Materials and Methods

After the literature review, the following relationships are expected. It is assumed that personal characteristics, mobility characteristics, and social neighborhood characteristics influence the use of public space, life satisfaction, and loneliness. In addition, the use of public spaces is expected to affect loneliness and life satisfaction. Finally, it is hypothesized that loneliness affects life satisfaction, meaning that people who feel lonely are less likely to report happiness or satisfaction. Figure 1 shows the expected relationships in a conceptual model.

### 3.1. Measurement

#### 3.1.1. Personal Characteristics

Participants were asked about their age, gender, income, educational background, household composition, and activities of daily living. Three categories were used to measure age: 18 to 35 years, 36 to 55 years, and 56 years or older. Income was measured with the categories low (lower than €1200 net per month), middle (between €1200 and €2400 net per month), and high income (higher than €2400 net per month), based on the net modal income of €2120 in the Netherlands. Educational background was broken into three categories, namely low, middle, and high education. Respondents with low education did not finish any education or finished primary or secondary school. Participants with middle education finished an intermediate vocational education. High educated participants finished a higher vocational or professional education. Household composition was measured by the categories single-person households, couples with children, and couples without children. The ethnic background of participants is not included as a variable, since the variation of ethnicity in the sample was extremely low.

Finally, activities of daily living (ADL) were measured using the Groningen Activity Restriction Scale (GARS), which consists of eighteen activities that show the ability or inability of people to perform activities of daily living [52]. The sum score of the eighteen activities can be used as a measure of ADL and can range from 18 to 72, since Cronbach’s Alpha (α) equals 0.955. A score of 18 indicates respondents who are not disabled and 72 represents people who are severely disabled.

#### 3.1.2. Mobility

Mobility characteristics that were included in the questionnaire were transport mode and walkability. The frequency of use of transport modes was operationalized by asking participants to indicate how often they use each of the transport modes on a seven-point scale, ranging from never to (almost) daily [15]. The categories walking and cycling are not included as transport modes, since these transport modes were included in the variables of public-space use. Next, walkability was included, measured by the Neighborhood Environment Walkability Scale, of which four items with the highest factor loadings were selected [53]. These items are: ‘Stores are within easy walking distance at my home’, ‘There are many alternative routes for getting from place to place in my neighborhood’, ‘The sidewalks in my neighborhood are well maintained’, and ‘There are many attractive natural sights in my neighborhood’. A sum score was calculated, which ranges from 4 to 20 (α = 0.687), where 4 indicates neighborhoods that are not perceived to be walkable and 20 indicates highly walkable neighborhoods.

#### 3.1.3. Social Neighborhood Characteristics

Two social neighborhood characteristics were addressed in the questionnaire; neighborhood attachment and social cohesion. Neighborhood attachment was measured by the six-item neighborhood attachment scale of Bonaiuto [54]. This scale consists of four aspects describing the bonds toward the neighborhood and people in the neighborhood. The other items are related to the participant’s desire to change or not to change their neighborhood. Some items are reverse-scoring items and should therefore be recoded. After recoding, the six items are summed (α = 0.848) and scores range from 6 to 30, where 6 indicates participants who are not attached and 30 indicates people who are very attached to their neighborhood.

The second subjective neighborhood characteristic that was included in the questionnaire was social cohesion. Social cohesion was measured by the social cohesion index of Frieling [55]. The index consists of seven items that are related to cooperation in the development of welfare, solidarity, and feelings of involvement. The seven items are again summed (α = 0.742), with a score ranging from 7 to 35. A score of 7 indicates neighborhoods that are not socially cohesive, and 35 indicates neighborhoods that were reported to have high social cohesion.

#### 3.1.4. Use of Public Space

Use of public space is measured by three questions in the questionnaire, to determine the patterns of outdoor-space use. First, participants were asked to indicate how frequently they walk in the neighborhood for different purposes. Second, respondents had to indicate how frequently they use specific public spaces. Third, participants were asked to answer how often they do specific activities in public space [56]. Each of these activities was measured on a seven-point scale, ranging from never to (almost) daily. The categorization of specific public spaces—parks, sport fields, community garden, day recreational area, agricultural area, and forest—is based on the categorization of land use in the Netherlands, which is composed by Centraal Bureau voor de Statistiek (CBS), Statistics Netherlands [57]. A principal component analysis (PCA) was performed to aggregate the activities to six components. The six components that arose after PCA were ‘recreational use’, ‘purposeful use and cycling’, ‘gardening’, ‘active use’, ‘passive use’, and ‘visit green space’. In the Appendix A, the components are described.

#### 3.1.5. Loneliness

Participants of the questionnaire were asked to indicate how often they feel lonely in their neighborhood. For this question, a three-item loneliness scale was used, which consists of three items and three answer possibilities; ‘hardly ever’, ‘some of the time’, and ‘often’. The three items were ‘How often do you feel that you lack companionship?’, ‘How often do you feel left out?’ and ‘How often do you feel isolated from others?’ [58]. Again, the sum of the three items was calculated (α = 0.843), and values ranged from 3 to 9, where 3 indicated respondents who do not feel lonely and 9 indicated people who often feel lonely.

#### 3.1.6. Life Satisfaction

Finally, people were asked to indicate how satisfied they were with their life. The Satisfaction with Life Scale was used to measure this variable [59]. The scale consists of five statements that can be answered by five answer categories, ranging from completely disagree to completely agree. A sum score was calculated that ranged from 5 to 25 (α = 0.864). Here, 5 indicated participants of the questionnaire who are not satisfied with their life, and 25 indicated people who are very satisfied with their life.

### 3.2. Data Collection

To study the relationships that were shown in Figure 1, data were collected using a cross-sectional approach, which consisted of an online questionnaire, including questions about demographics, frequency of use of public spaces, the perception of neighborhood characteristics, and mobility patterns. The questionnaire was approved by an ethics committee, and respondents agreed to give informed written consent to participate in the survey. Data were collected in June and July 2019, among the residents of the three selected neighborhoods in the middle-sized Dutch city ‘s-Hertogenbosch. The link to the web-based questionnaire of the research was distributed via social media. Meetings with community centers and elderly centers were conducted to increase the response of the elderly target group in the three neighborhoods of ‘s-Hertogenbosch. After two months, 297 residents responded to the questionnaire, of which 200 responses could be used in further analyses. Ninety-seven cases were deleted, since these participants did not indicate they lived in any of the three neighborhoods. Fifty-three percent of the participants indicated they lived in Maaspoort, 21% in the Binnenstad (inner city), and 26% in Rosmalen Zuid.

The city ‘s-Hertogenbosch has a population of almost 155,000 (January 2019), compared to a population of nearly 863,000 in Amsterdam (January 2019), which is the biggest city and the capital of the Netherlands. The highway A2 connects the cities Amsterdam, Utrecht, ‘s-Hertogenbosch, and Eindhoven. The left image of Figure 2 indicates the locations of ‘s-Hertogenbosch, Utrecht, Amsterdam, and Eindhoven. The zoomed image shows ‘s-Hertogenbosch, with the three selected neighborhoods for data collection: Maaspoort, Binnenstad (inner city), and Rosmalen Zuid. These neighborhoods were selected for their differences in population and physical characteristics. As is shown in Table 1, the inner city is characterized by a high percentage of facilities and services, which are situated around the main, medieval square of the city. Binnenstad is also located within walking distance from the central train and bus station and is well connected to surrounding neighborhoods. In Maaspoort, the number of facilities, such as shops and restaurants, is lower, but the amount of land that is covered with parks and recreational areas is relatively high. Rosmalen Zuid is characterized by the highest percentage of elderly people but a lower number of facilities. Rosmalen Zuid has its own train station and is therefore well connected by train.

### 3.3. Analytical Approach

The analytical approach that was chosen to analyze the effects of personal characteristics, mobility, and social neighborhood characteristics on public-space use, loneliness, and life satisfaction, was path analysis. The advantage of using a path analysis over bivariate or regression analysis is that multiple, both direct and indirect, relationships between independent and dependent variables can be tested simultaneously. Path analysis is a special case of structural equation modelling, in which only observed variables are used [60]. The path model was estimated using the statistical package LISREL version 8.54.

Before building the path model in LISREL, bivariate analyses between the independent and dependent variables, as indicated in the conceptual model (see Figure 1), were performed to test the significance of hypothesized relationships. All significant relationships of the bivariate analyses were added to the path model. To reduce the number of variables in the path analysis and to overcome the risk of overfitting the model, the relationships that were not found to be significant at the 0.1 (t ≥ 1.65) significance level in the path model were removed. The backward stepwise process was repeated until an acceptable model fit was found and all insignificant relationships were removed from the model.

## 4. Results

### 4.1. Sample Description and Personal Characteristics

Table 2 describes the personal characteristics of the sample. The sample is compared to nationwide averages, measured by CBS [61] and to city-specific data of ‘s-Hertogenbosch, collected by the municipality of the city [62]. The table shows that the sample is not entirely representative of the population of ‘s-Hertogenbosch or the Netherlands. The oldest age group in the research is purposefully overrepresented, whereas the middle-aged participants are underrepresented. The sample also contains many female participants and residents with a high educational background. Finally, it can be observed that people who live in households without children are somewhat overrepresented.

In addition to nominal variables, Table 2 also presents the mean and standard deviations of activities of daily living. The mean of the activities of daily living of participants of the research equals 21.88, which indicates that the independence of the sample is high.

Table 3 shows the mean values and standard deviations of the mobility characteristics, neighborhood characteristics, use of public space, loneliness, and life satisfaction. As is shown in the table, participants use the car (as a passenger or as a driver) more often than public transport (trains and buses). For the perception of walkability, a mean value was found that indicates that walkability in the three neighborhoods of ‘s-Hertogenbosch is perceived to be average. The mean value of neighborhood attachment equals 18.873, which indicates that most participants feel somewhat attached to their neighborhood. For social cohesion, a mean of 22.289 was found, showing that participants reported social cohesion to be average. In general, participants do not feel lonely and are quite satisfied with their life.

### 4.2. Path Analysis

Table 4 shows the goodness of fit of the model. Chi-Square divided by the degrees of freedom should ideally be smaller than 2, but smaller than 5 is acceptable [63]. In the current research, Chi-Square is 2.5 times larger than the degrees of freedom. The root mean square error of approximation (RMESA) should, additionally, be below 0.05, in order to indicate a good fit of the model to the data [64]. The RMSEA of the current model equals 0.089, which indicates a modest fit. The Goodness of Fit Index equals 0.91 and should preferably be greater than 0.90 to indicate a good model [64].

### 4.3. Relationships Path Model

With the results of the path model, the expected relationships between the dependent variables and the explanatory variables could be tested. Table 5 and Figure 3 indicate the significant relationships in the path model at the 0.1 significance level. In Figure 3, significant positive relationships are drawn with a black arrow and negative relationships with a dashed black arrow. Since no significant relationships are found between any of the explanatory variables and gardening, this component is deleted from further analysis and from the path model.

#### 4.3.1. Recreational Use

Table 5 shows that age, activities of daily living (ADL), neighborhood, and neighborhood attachment significantly affect recreational use. Positive relationships are found between the two oldest age groups and recreational use. This indicates that residents who are older use public spaces for recreational purposes more often. In addition,, a significant relationship between ADL and recreational space use is found. People with a lower ADL use public spaces for recreational activities less often than people with a high ADL. Recreational use of public spaces is also found to be positively affected by the neighborhoods Binnenstad and Maaspoort. Residents of these neighborhoods more frequently use public spaces for recreational purposes than residents of Rosmalen Zuid. Finally, neighborhood attachment affects recreational use. People who are attached to their neighborhood use public spaces for recreational activities more often.

#### 4.3.2. Purposeful Use and Cycling

Purposeful use and cycling are significantly affected by the variables train use, neighborhood, social cohesion, and perception of walkability. People who frequently use the train are also more likely to frequently use public spaces for specific purposes or to cycle. Residents of the neighborhoods Maaspoort and Binnenstad use public spaces for specific purposes or for cycling more often than residents of Rosmalen Zuid. Purposeful use and cycling are also affected by social cohesion, which indicates that people who live in socially cohesive neighborhoods cycle or use public space for specific purposes more often than other residents. Finally, people who observe their neighborhood to be walkable purposefully use public spaces or cycle in their neighborhood more often.

#### 4.3.3. Active Use

For active use, the variables education, train use, neighborhood, and perception of walkability are found to be influential. Residents with a high educational background use public space for active purposes more frequently than other residents. Moreover, frequent train users are more likely to actively use public spaces. Residents of the neighborhood Binnenstad also use public spaces for active purposes more often than residents of Rosmalen Zuid and Maaspoort. Furthermore, the perception of walkability significantly affects active use, indicating that residents who perceive their neighborhood to be walkable use public space for active purposes more often.

#### 4.3.4. Passive Use

Passive use is significantly affected by activities of daily living (ADL), car use as a driver, neighborhood, social cohesion, and neighborhood attachment. ADL is negatively related to passive space use, which indicates that people who are more dependent on others in performing ADL use public spaces for passive activities less frequently. In addition, people who frequently drive a car are found to use public spaces for passive activities less frequently. Furthermore, residents of the neighborhood Binnenstad more often passively use public spaces than residents of Maaspoort and Rosmalen Zuid. Social cohesion is also found to be associated with passive use, indicating that residents of socially cohesive neighborhoods visit public spaces for passive activities more often than residents of neighborhoods with weaker social cohesion. Finally, residents who are attached to their neighborhood use public spaces for passive activities more often.

#### 4.3.5. Visiting Green Spaces

Income, household composition, neighborhood attachment, social cohesion, and perception of walkability all significantly affect the visiting of green spaces. Residents with a higher income visit green spaces more frequently than people with a low or moderate income. Couples with children and couples without children visit green spaces more often than single-person households. Next, residents who are attached to their neighborhood visit green spaces more frequently than residents who do not feel attached. Social cohesion, in addition, also affects the frequency of visits to green spaces, indicating that residents who live in socially cohesive neighborhoods visit green spaces more often than other residents. Finally, people who perceive their neighborhood to be walkable visit green spaces more often.

#### 4.3.6. Loneliness

Next to the relationships with the five components of public-space use, significant associations with loneliness are found in the path analysis. The variables household composition, activities of daily living (ADL), social cohesion, and passive space use are all observed to significantly affect the loneliness of people. Single persons are found to be lonelier than couples with or without children. Moreover, residents who are dependent on others in performing ADL generally feel lonelier than people who are not dependent on others. Another significant relationship exists between social cohesion and loneliness: People who report social cohesion to be low feel lonely more often than residents of socially cohesive neighborhoods. Finally, passive space use affects loneliness, which indicates that people who frequently use public space for passive activities are less likely to feel lonely. Only one of the five components of public-space use is thus significantly associated with loneliness.

#### 4.3.7. Life Satisfaction

For life satisfaction, significant effects of activities of daily living (ADL), social cohesion, active use, and loneliness are observed. Residents who can perform ADL independently are more likely to be satisfied with their life. Moreover, people who live in socially cohesive neighborhoods are more likely to report high life satisfaction. Next, active use affects life satisfaction, which indicates that residents who frequently visit public spaces for active purposes are more likely to be satisfied with their life. Only one of the five components of public-space use significantly affects life satisfaction. Finally, a significant effect of loneliness on life satisfaction can be observed, which indicates that people who feel lonely are less satisfied with their life.

## 5. Discussion

The main aim of the current research was to analyze the relationships between personal, mobility, and social neighborhood characteristics and public-space use, loneliness, and life satisfaction. In addition, the effect of public-space use on loneliness and life satisfaction, and the effect of loneliness on life satisfaction, was studied. The results showed a significant relationship between loneliness and life satisfaction, but a limited effect of public-space use on loneliness and life satisfaction was observed. For life satisfaction, a relationship with active use was found. In addition, for loneliness, a relationship with passive use was found. The effect of passive use on loneliness indicates that the enjoyment of public-space visits is more important than the actual activities that are performed in the public space. Based on data collected using a questionnaire that was distributed via social media and meetings with community and elderly centers, 200 responses were retrieved, and a path model was estimated. As was shown, the RMSEA of the path model equaled 0.089, which is rather high. This may be caused by the large number of paths relative to a small sample (N = 200) [65].

Due to the nonrandom distribution method of the questionnaire, by visiting community and elderly care centers, purposeful overrepresentation of the oldest age group occurred. Differences between the population of ‘s-Hertogenbosch and the sample were found for gender. In the sample, females were overrepresented, which may be a result of the time that women spend at home relative to the time spend by males. Overrepresentation of people with a high educational background may, in addition, be caused by the willingness of respondents to participate in the study, which is generally higher among highly educated individuals. Finally, the overrepresentation of households without children may be related to the higher percentage of residents aged above 56, who usually have children that moved out of house.

As was shown in the path analysis, residents above the age of 56 visit public spaces for recreational activities more frequently than people aged between 35 and 55. Previous research explained this finding by arguing that middle-aged residents are more often employed full-time and travel outside their neighborhood on a daily basis. They are, therefore, also more likely to use public spaces outside their neighborhood [66]. We did, however, not find a significant effect of age on life satisfaction or loneliness, while other studies did find this effect [38,50]. An ANOVA test (F = 6.007, *p*-value = 0.003) indicated that the dependence on others to perform activities of daily living (GARS score) increases with age (for 18–35 years old M = 19.06, SD = 4.388, for 36–55 years old M = 20.81, SD = 8.55, for 56 years or older M = 24.04, SD = 10.37). Differences in loneliness and life satisfaction between age groups can thus be explained by differences in ADL.

ADL was also found to affect recreational use and passive use. Although these relationships have not been studied before, previous studies did show that people with limited ADL walk significantly less for recreational activities than residents without functional limitations [67]. Residents with a low ADL have lower mobility levels, which further limits the options to visit public spaces [68,69]. These residents are therefore more likely to perceive neighborhood facilities to be poor [68]. Due to physical limitations, residents are probably not able to reach public spaces independently and, therefore, cannot use them in any form.

Other personal characteristics that were found to affect active use or green-space visitation were education and household composition. Previous studies showed that highly educated residents are more likely to use public spaces for family and sporting activities than lower educated people [70]. In general, those with a higher education are more likely to be physically active according to the prevailing recommendations than those who did not complete secondary school [71]. In addition, young families with children, or families who expect to have children, move to family houses that are close to nature, to be able to use public spaces [70,72]. Public spaces can provide opportunities for children to play and for parents and other adults to meet with neighbors. The finding that single persons are lonelier than couples with or without children may also be related to the social interactions in public spaces. More in-depth research is needed on the actual relationships between household composition, income, education, and public-space use and loneliness.

Next to the personal characteristics, the perception of walkability was found to significantly affect active use, purposeful use and cycling, and the visiting of green spaces. Residents who perceived their neighborhood to be unwalkable were less likely to use public spaces. These findings are confirmed by a previous study in which a lack of walkability emerged as one of the main barriers to use public spaces [56]. In contrast, people who live in highly walkable neighborhoods cycle and walk to public spaces more frequently and can enhance their health [73]. More specifically, users of parks and other green spaces have higher daily walking minutes than people who do not use green spaces [74]. The walkability of neighborhoods encourages residents to go outside [75] and to socially engage in their living environment [41]. Highly walkable, green neighborhoods, moreover, prevent residents from becoming lonely. Policymakers should therefore aim to create neighborhoods that are highly walkable, where people are invited to go outside and meet their neighbors. Walkability can be promoted by a higher residential density, mixed land use, and high street connectivity [75]. In the case of ‘s-Hertogenbosch, the presence of accessible, natural features, such as parks and green spaces, that are well connected to the residential areas, can promote walkability. These neighborhood features may also contribute to the satisfaction of all residents and may prevent dependent residents from becoming lonely.

Th results of the current study also indicate that frequent train users are more likely to use public spaces for specific purposes and cycle more than others in the neighborhood. Moreover, train users frequently use public spaces for active purposes. Both these results indicate that public-transport users are more physically active than people who use other transport modes, such as a car. Previous theory has shown that the mobility characteristics of an individual affect the satisfaction with life [10] and loneliness [21]. People who have access to different transport modes, such as the car and public transport, were found to have more social interactions with others [21]. Moreover, public-transport users meet physical activity recommendations more easily, since they walk significantly more than car users. As was also shown for users of green spaces, public transport users were found to have longer daily walks than people who use the car [74]. Policymakers should therefore focus on creating public-transport facilities that are highly accessible, to promote both public transport and active transport modes within urbanized areas. If public transport facilities are within walking distance, residents may be more likely to use them and to reduce their car use.

Finally, neighborhood attachment and social cohesion have a significant effect on the use of public spaces. Neighborhood attachment was found to affect recreational use, passive use, and the visiting of green spaces. Social cohesion was also found to affect passive use, the visiting of green space, and purposeful use and cycling. Moreover, social cohesion significantly affected loneliness and life satisfaction. To the best of our knowledge, the direct effects of neighborhood attachment and social cohesion on the use of public space have not been studied before. Previous research mainly focused on the indirect relationship between neighborhood attachment or social cohesion and public-space use, through the effect of social interactions in a neighborhood [71,76]. Zhang and Zhang (2017), for instance, argued that people feel more attached to their neighborhood when opportunities to communicate with others exist in public spaces. In these socially cohesive neighborhoods, people are generally less prone to feeling lonely [45]. To reduce loneliness and increase life satisfaction of residents in a neighborhood, social cohesion should be promoted. In socially cohesive neighborhoods, public spaces encourage residents to meet neighbors and to have social interactions, thereby strengthening feelings of attachment and social cohesion. Policymakers should therefore strive to create public spaces that are inviting, where people can meet with neighbors and feel socially involved. Future research could look in more detail at the relationships between social cohesion, public-space use, and life satisfaction and loneliness for urban, as well as rural, neighborhoods.

## 6. Limitations

One of the limitations of this research is that it was based on a rather small sample of ‘s-Hertogenbosch in the Netherlands. Although elderly participants of the survey were recruited via meetings with community and elderly centers, other participants of the survey could respond via social media. These respondents may therefore be more active, both physically and psychologically, in the neighborhoods than residents who did not participate. Moreover, an overrepresentation of highly educated and high-income participants occurred in the sample. Furthermore, no racial variation existed, and, therefore, ethnic background was deleted from the analysis. In addition, controlling for ADL is important; however, the measurement of ADL is usually used for frail older adults, while this sample contains a relatively large share of younger adults. Results of the study should therefore be carefully interpreted. Relationships between the three selected neighborhoods and public-space use are also significant, which indicates that differences between the neighborhoods are influential. It is, therefore, not possible to generalize the outcomes to other cities of the Netherlands or cities outside the Netherlands.

For future research, it would be interesting to analyze the use of public spaces, loneliness, and life satisfaction of people living in different regions and countries, to find whether cultural differences occur. Future research could also include more extensive characteristics of public space, such as the observed quality and distance to public space, to get a more comprehensive understanding of public-space use on life satisfaction and loneliness. Finally, future research should use a larger sample, including more neighborhoods and other regions, to reduce the RMSEA of the path analysis, to ensure a good model fit to the data, without losing significant paths. Performing the study in other regions and in more neighborhoods of a city also helps to gain more insights into the relationships that were significant in the current study.

## 7. Conclusions

Hitherto, research that simultaneously analyzes the effects of both personal and social neighborhood characteristics on life satisfaction and loneliness is still limited [19]. Therefore, the main contribution of this study is that the relationships between personal, neighborhood, and mobility characteristics and life satisfaction and loneliness are observed in an empirical study, whereas previous studies mainly focused on only one of these characteristics. This study, moreover, analyzed the frequency with which residents use public spaces and whether public-space use affects loneliness or life satisfaction. A path analysis was used to analyze the significance of these relationships simultaneously. The questionnaire that was used to obtain the data was distributed via social media and community centers, enabling a diverse group of residents of ‘s-Hertogenbosch to participate. As a result, a heterogenous sample of residents of three neighborhoods of ‘s-Hertogenbosch was retrieved.

The results showed that personal, neighborhood, and mobility characteristics influence specific uses of public spaces, loneliness, and life satisfaction. Loneliness is also found to affect life satisfaction, indicating that residents who feel lonely are less likely to be satisfied with their life. The relationships between public-space uses and loneliness and life satisfaction are, however, found to be limited. As was also argued in previous research, the effect of age on public-space use, loneliness, and life satisfaction is small; instead, it was found that the dependence of people in performing activities of daily living is influential. In general, ageing residents are most dependent on others in performing daily activities.

Overall, results of this study are relevant for policymakers who focus on creating cohesive, walkable, and accessible neighborhoods that contribute to the well-being of all its residents. Specifically, walkability and accessibility of public space, supported by public-transport facilities and green spaces, should be promoted to support physical activity and independence of residents of all age groups.

## Figures and Tables

**Figure 1 ijerph-16-04282-f001:**
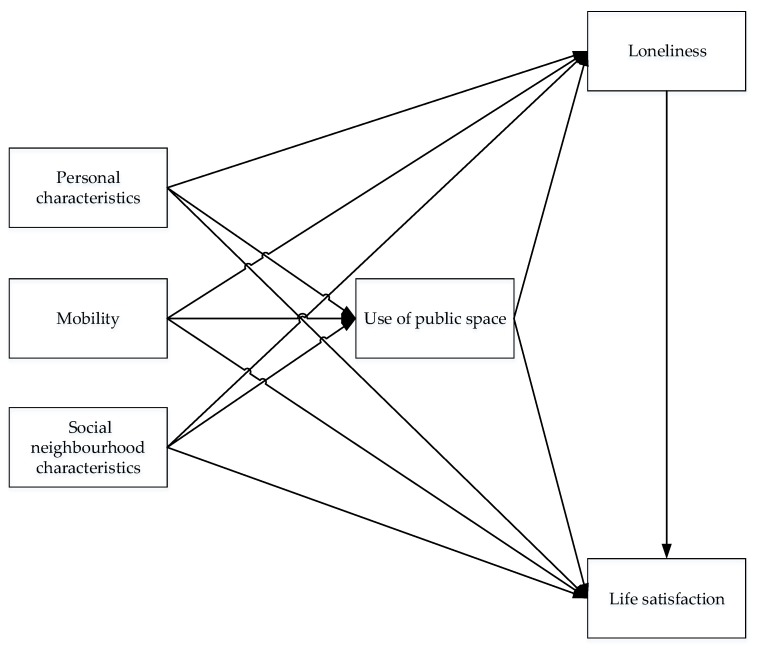
Conceptual model.

**Figure 2 ijerph-16-04282-f002:**
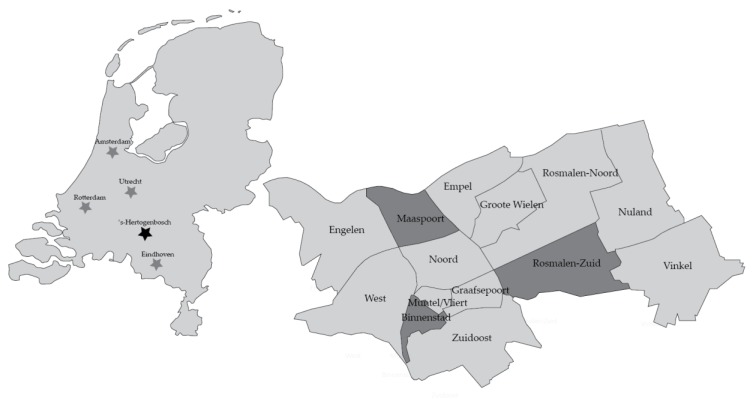
Three selected neighborhoods in ‘s-Hertogenbosch, the Netherlands.

**Figure 3 ijerph-16-04282-f003:**
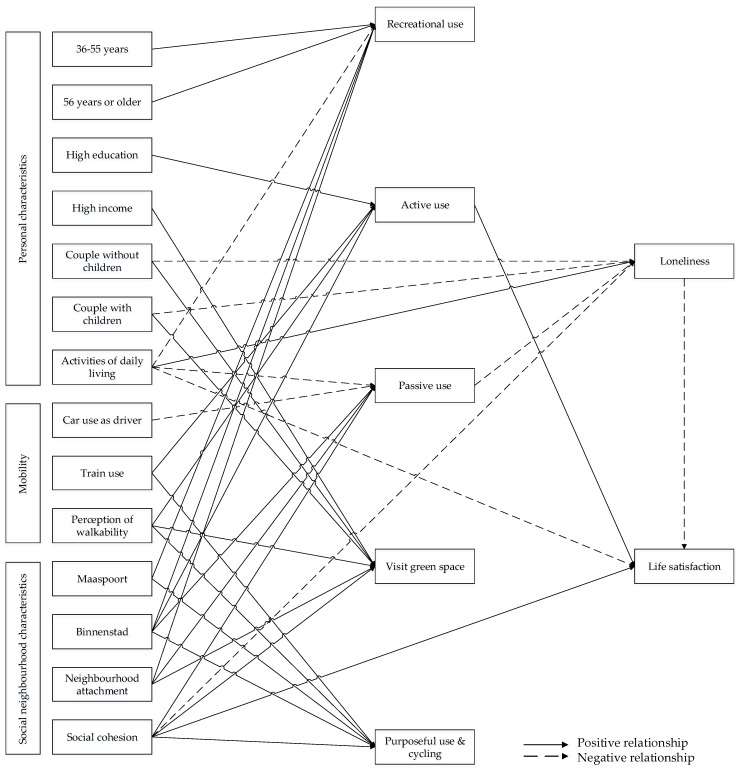
Relationships in path model.

**Table 1 ijerph-16-04282-t001:** Neighborhood characteristics.

	Maaspoort	Binnenstad	Rosmalen Zuid
*Demographics*			
Population size	16,734	12,840	9329
Age above 56 (%)	31.9	29.7	40.1
*Retail*			
Distance to big supermarket (km)	0.7	0.4	1.3
Nr. within 1 km	1.3	3.2	0.8
*Restaurants*			
Distance to restaurant (km)	0.7	0.9	0.2
Nr. within 1 km	1.2	96.8	1.9
*Transport*			
Distance to train station (km)	4.3	1.1	1.8
*Parks and green areas*			
Total area (ha)	565	236	1043
Recreational and parks (%)	22.3	1.6	6.9
*Maintenance*			
Average rate (1–10)	7.3	7.4	7.1

**Table 2 ijerph-16-04282-t002:** Characteristics of participants (N = 200).

	Sample (%)	‘s-Hertogenbosch (%)	Netherlands (%)
*Age*			
18–35 years	25.5	25.5	24.3
36–55 years	27.5	35.3	33.8
56 years or older	47	39.2	41.9
*Gender*			
Male	27.5	49	49.7
Female	72.5	51	50.3
*Education*			
Low	20.5	28	31.1
Moderate	24.5	37.5	37.8
High	55	34.5	31.1
*Income*			
Low	22.5		
Moderate	41.5		
High	29.5		
Don’t know	6.5		
*Household composition*			
One-person household	28	38.2	39.7
Household without children	36	28.8	28.7
Household with children	36	33	31.6
	**Mean**	**St. deviation**	
Activities of daily living (19–72)	21.88	8.915	

**Table 3 ijerph-16-04282-t003:** Social neighborhood characteristics, mobility, loneliness, and life satisfaction (N = 200).

	Mean	St. Deviation
Car use as a passenger (1–7)	4.32	1.89
Car use as a driver (1–7)	5.29	2.07
Bus use (1–7)	2.15	1.53
Train use (1–7)	2.41	1.62
Perception of walkability (4–20)	15.04	3.19
Neighborhood attachment (7–30)	18.87	4.85
Social cohesion (7–32)	22.29	5.06
Frequency of walking/cycling (8–56)	27.90	8.31
Frequency of use of specific spaces (7–49)	14.59	6.60
Frequency of specific activities (8–56)	22.65	6.35
Loneliness (3–9)	3.82	1.36
Life satisfaction (5–25)	18.29	4.04

**Table 4 ijerph-16-04282-t004:** Goodness-of-fit of the model.

Degrees of Freedom	100
Full Information ML Chi-Square	254.02
RMSEA (Root Mean Square Error of Approximation)	0.089
90 Percent Confidence Interval for RMSEA	0.075; 0.10
*P*-value for Test of Close Fit (RMSEA < 0.05)	0.0000
Goodness of Fit Index	0.91

**Table 5 ijerph-16-04282-t005:** Results path model (unstandardized coefficients).

		Life Satisfaction	Loneliness	Recreational Use	Purposeful Use and Cycling	Active Use	Passive Use	Visit Green Space
*Variables*	*Categories*	*Unstandardized coefficients*						
Age	18–35	-	-	-	-	-	-	-
	36–55			4.31 **				
	56 or above			3.39 **				
Education	Low					−0.63		
	Moderate	-	-	-	-	-	-	-
	High					1.02 *		
Income	Low							0.65
	Moderate	-	-	-	-	-	-	-
	High							1.00 **
	Unknown	-	-	-	-	-	-	-
Household composition	Single persons	-	-	-	-	-	-	-
	Couple without children		−0.87 **					0.88 *
	Couple with children		−0.55 **					1.04 **
Activities of daily living		−0.07 **	0.03 **	−0.18 **			−0.10 **	
Car use as a driver							−0.48 **	
Train use					0.82 *	0.62 **		
Neighborhood	Maaspoort			4.68 **	1.75 *	−0.46	−1.16	
	Binnenstad			5.69 **	4.75 **	3.40 **	1.58 *	
	Rosmalen Zuid	-	-	-	-	-	-	-
Neighborhood attachment				0.38 **			0.14 **	0.07 *
Social cohesion		0.12 **	−0.04 **		0.27 **		0.18 **	0.08 **
Perception of walkability					0.48 **	0.25 **		0.11 *
Active use		0.10 *						
Passive use			−0.04 *					
Loneliness		−0.93 **						
**R-Squared**		**0.22**	**0.19**	**0.20**	**0.29**	**0.39**	**0.23**	**0.16**

** *p* < 0.05, * *p* < 0.1.

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
