# Peer review of "Loneliness and Life Satisfaction Explained by Public-Space Use and Mobility Patterns"

_ijerph, 2019, doi:10.3390/ijerph16214282_

Round 1

Reviewer 1 Report

This study examines the public space use with individual and neighborhood factors associated with loneliness and life satisfaction in Netherlands. The contextual effect to psychological well-being has been recently explored, and this study adds more evidence. Furthermore, this study also used PCA to extract difference use of public space, and use LISREL to examine the possible causal relationships among variables. In general, this study is recommended. Here are some minor comments and suggestions.

The participants were recruited through social media and community meetings. It is possible that the participants were more physically and psychologically active than the whole population, as the result of Table 1. This point may be added in the limitations. The sample was purposely selected from several neighborhoods. Although the authors showed the map and gave some description about the differences in the neighborhoods, the readers not from Netherlands may not aware of the context. Please add more descriptions about the study areas to be urban or rural, especially about the transportation accessibility. ADLs were used to measure the physical function, which is usually used for frail older adults. The sample of this study was not only from the older adults but also from younger adults. Thus, the measure of ADLs may not be a sensitive measurement of physical function or mobility ability. This may be discussed in the 2nd paragraph of the Discussion about ADLs. However, to controlling for ADLs in the model is still necessary, because it is highly related to the ability to use public space. Passive use was found to be negatively related to loneliness. It seems that passive use is in the middle of something, such as the way to other activities or event. It can also be like an enjoyment of the outdoors. In addition, social cohesion was related to both loneliness and life satisfaction and three kinds of public space use. More discussion about social cohesion is suggested. In general, the discussion of the results are written well. Usually the explanation of the results will be discussed in the Discussion section, but this manuscript is in a different way, that puts the explanations in the Result section. I suggest the Result section only describe the findings, but the explanations and the discussions of the results can be moved to the Discussion section.

Author Response

This study examines the public space use with individual and neighborhood factors associated with loneliness and life satisfaction in Netherlands. The contextual effect to psychological well-being has been recently explored, and this study adds more evidence. Furthermore, this study also used PCA to extract difference use of public space, and use LISREL to examine the possible causal relationships among variables. In general, this study is recommended. Here are some minor comments and suggestions.

Thank you for your supportive words and your valuable comments. Based on your suggestions, revisions to the paper have been made.

The participants were recruited through social media and community meetings. It is possible that the participants were more physically and psychologically active than the whole population, as the result of Table 1. This point may be added in the limitations.

Thank you for pointing out this limitation. The comment has been added to the limitations section.

The sample was purposely selected from several neighborhoods. Although the authors showed the map and gave some description about the differences in the neighborhoods, the readers not from Netherlands may not aware of the context. Please add more descriptions about the study areas to be urban or rural, especially about the transportation accessibility.

Thank you for raising this issue. A more comprehensive description of both the city and its neighbourhoods is added.

ADLs were used to measure the physical function, which is usually used for frail older adults. The sample of this study was not only from the older adults but also from younger adults. Thus, the measure of ADLs may not be a sensitive measurement of physical function or mobility ability. This may be discussed in the 2nd paragraph of the Discussion about ADLs. However, to controlling for ADLs in the model is still necessary, because it is highly related to the ability to use public space.

Thank you for this comment. We added a line in the limitations section that describes a possible limitation caused by using the GARS measurement method.

Passive use was found to be negatively related to loneliness. It seems that passive use is in the middle of something, such as the way to other activities or event. It can also be like an enjoyment of the outdoors.

Thank you. We added some extra explanation of this result in the discussion.

In addition, social cohesion was related to both loneliness and life satisfaction and three kinds of public space use. More discussion about social cohesion is suggested.

Thank you. We agree with your argument and therefore included a new paragraph in the discussion section.

In general, the discussion of the results are written well. Usually the explanation of the results will be discussed in the Discussion section, but this manuscript is in a different way, that puts the explanations in the Result section. I suggest the Result section only describe the findings, but the explanations and the discussions of the results can be moved to the Discussion section.

Thank you. We moved the explanation of the results to the discussion section, leaving the results in the result section.

Many thanks for your extensive review and positive feedback! We have carefully revised the paper, based on your comments.

Reviewer 2 Report

This paper aims to explore how public space use mediates the relationship of personal, neighbourhood and mobility characteristics with loneliness and life satisfaction in an urban setting. The topic is of utmost importance in light of the evidence that within the next three decades, almost 70% of the world’s population is expected to live in urban areas. The study was conducted on a convenience sample in a middle-size Dutch city. Yet the authors admit the sampling method and sample size compromise the ability to generalize their findings. The manuscript is well written and well organized. Data analysis by means of structural equation modelling seems to be a great choice to work with mediations and indirect effects. I just have a few minor comments which are thought to add to the paper:

Abstract >> What does ADL stand for?

Methods >> The number of participants must be reported in this section. Also, the share of participants with complete data which is the final sample size.

Methods >> Do the authors obtained approval from an ethics committee? If so, this must be reported in this section of the manuscript. Also, the authors should report whether participants signed informed written consent.

Methods >> Please provide a range of income for each income category. Providing this data in euros is fine.

Methods >> Please provide a specific definition for low, middle and high educational background for those of us who are not familiarized with the Dutch educational system.

Methods >> Please provide some information on the city of 's-Hertogenbosch, i.e. estimated population, closeness to major Dutch cities such as Amsterdam or Rotterdam, main economic activity, etc.

Discussion

Because the findings have the potential to inform policymaking I’d like to see more discussion on the policy implications.

Author Response

This paper aims to explore how public space use mediates the relationship of personal, neighbourhood and mobility characteristics with loneliness and life satisfaction in an urban setting. The topic is of utmost importance in light of the evidence that within the next three decades, almost 70% of the world’s population is expected to live in urban areas. The study was conducted on a convenience sample in a middle-size Dutch city. Yet the authors admit the sampling method and sample size compromise the ability to generalize their findings. The manuscript is well written and well organized. Data analysis by means of structural equation modelling seems to be a great choice to work with mediations and indirect effects. I just have a few minor comments which are thought to add to the paper:

Thank you for your positive words! Based on your suggestions, we revised the paper.

Abstract >> What does ADL stand for?

Thank you, we included the written definition of ADL in the abstract.

Methods >> The number of participants must be reported in this section. Also, the share of participants with complete data which is the final sample size.

Thank you. We already included the number of participants and the final sample size in this section but moved it forward.

Methods >> Do the authors obtained approval from an ethics committee? If so, this must be reported in this section of the manuscript. Also, the authors should report whether participants signed informed written consent.

Thank you for this comment. An ethics committee of our University approved the questionnaire. A informed written consent was also added to the website where participants could respond. They did not have to sign the informed written consent.

Methods >> Please provide a range of income for each income category. Providing this data in euros is fine.

Thank you for this suggestion. We added the income ranges for the three categories.

Methods >> Please provide a specific definition for low, middle and high educational background for those of us who are not familiarized with the Dutch educational system.

Thank you. We added a more specific definition of the three educational backgrounds with a comparison with the American educational system.

Methods >> Please provide some information on the city of 's-Hertogenbosch, i.e. estimated population, closeness to major Dutch cities such as Amsterdam or Rotterdam, main economic activity, etc.

Thank you for this comment. We added a more comprehensive description of both the city ‘s-Hertogenbosch as well as of the three selection neighbourhoods.

Discussion

Because the findings have the potential to inform policymaking I’d like to see more discussion on the policy implications.

Thank you for your suggestion. We added some extra discussion points and policy implications.

Many thanks for your valuable suggestions! We revised our paper based on your thorough review.

Reviewer 3 Report

Overall Comments:

This paper provides a comprehensive examination of personal, mobility and neighborhood factors that influence an individual’s feelings of loneliness and life satisfaction. Additionally, this manuscript examines an elderly population, one that is rapidly growing and may greatly benefit from changes to the environment. The insights from this study are applicable and important for bridging the research- policy gap.

Overall, authors did a great job. The manuscript provides a breadth of literature in the introduction, literature review and discussion. The authors provide detail and transparency in their methods. My main concerns are regarding the organization of the manuscript- particularly the presentation of results. Currently, authors discuss the results within the results section- these should be separated. Overall, the authors should work to use concise language and remove repetition throughout the manuscript.

Abstract

ADL is not previously defined.

Introduction and Literature Review

The introduction and literature review are very comprehensive. This amount of literature is more than I am used to seeing in a manuscript. However, the authors did a nice job of setting up the research question and summarizing the literature.

Lines 67-74 can be/should be removed from the introduction. Lines 67-70 are methods and are already presented later in the manuscript. Lines 71-74 are unnecessary.

The first part of the literature review Lines 76-96 should be compared to the introduction to eliminate repetition.

Section 2.1 Personal Characteristics- does not discuss race? Has this been shown to affect life satisfaction and loneliness? In general, I think the exclusion of this should be justified throughout the manuscript.

Materials and Methods

Lines 173-175- delete sentence outlining the rest of the section. This is unnecessary.

Conceptual Model- why did you not explore how personal characteristics influence mobility and neighborhood characteristics? We know that individuals who are low SES or lower educational status are more likely to live in disadvantaged neighborhoods, which may be characterized as having low social cohesion and low walkability. Additionally, shouldn’t there a be a line from mobility to neighborhood characteristics? As you state in your introduction, an environment that is more walkable foster social interactions that in turn increase social cohesion. Therefore, you would hypothesize that mobility influences neighborhood characteristics.

3.1.2- which of the four walkability scale items were included in the factor?

3.1.3.- These are all social environmental measures. I may suggest terming these “Social Neighbourhood Characteristics.” The physical environment is actually captured within your mobility construct as walkability. There is a little bit of disconnect between your literature review and study because you make a case for the importance of physical factors in the neighborhood environment under the Neighbourhood Characteristics section (2.3- lines 160-166) but that is not measured in this domain or could be argued that it is measured under mobility.

3.1.4- is it not clear what was included after your principal component analysis from this text. The authors provide additional detail in the results section- however, these are methods. I would clarify briefly in the text here and add the details you provide in Lines 309-333 to an appendix.

3.2 Data Collection- You mention in later in the paper (section 4.1) that elderly populations were targeted, however that is not made clear in your data collection methods. Please clarify in this section whether elderly centers were targeted. These methods should appear here, then section 4.1 should present results only, the justification for why your study population is not representative of the Netherlands, etc should be in the discussion section.

Results

4.1- I love Table 1 and how it compares your population to that of the Netherlands and s-Hertogenbosch. These results should state the main differences between your study population and the other two population, but not discuss why this is the case. This discussion should be within your discussion section only.

Remove Lines 309-333 and make an appendix that is referenced within your methods section.

Table 2- did you not ask about active transport (e.g biking or walking?) You will need to justify this.

All results sections need to be majorly revised to simply state the results. The majority of your current results are actually discussion points and can be integrated into your current discussion.

Discussion

Again, I would move literature support from your results sections to the discussion to highlight your key findings and add to your discussion. Otherwise, I like the discussion and conclusion. The summary of findings in the discussion is concise and helps consolidate findings for the reader.

Should add consider adding the lack of examination of race to the limitations section. Additionally, how did a sample that was largely higher income and higher educational status influence findings?

Author Response

This paper provides a comprehensive examination of personal, mobility and neighborhood factors that influence an individual’s feelings of loneliness and life satisfaction. Additionally, this manuscript examines an elderly population, one that is rapidly growing and may greatly benefit from changes to the environment. The insights from this study are applicable and important for bridging the research- policy gap.

Overall, authors did a great job. The manuscript provides a breadth of literature in the introduction, literature review and discussion. The authors provide detail and transparency in their methods. My main concerns are regarding the organization of the manuscript- particularly the presentation of results. Currently, authors discuss the results within the results section- these should be separated. Overall, the authors should work to use concise language and remove repetition throughout the manuscript.

Many thanks for your thorough review of our paper. We revised and reorganised the paper based on your suggestions.

Abstract

ADL is not previously defined.

Thank you. We included the written definition of ADL in the abstract.

Introduction and Literature Review

The introduction and literature review are very comprehensive. This amount of literature is more than I am used to seeing in a manuscript. However, the authors did a nice job of setting up the research question and summarizing the literature.

Thank you!

Lines 67-74 can be/should be removed from the introduction.

Thank you, we removed these lines.  

Lines 67-70 are methods and are already presented later in the manuscript.

Thank you, we removed these lines.

Lines 71-74 are unnecessary.

Thank you, these lines are removed as well.

The first part of the literature review Lines 76-96 should be compared to the introduction to eliminate repetition.

Thank you for your comment. We compared the introduction and the literature review and rewrote it to eliminate repetition.

Section 2.1 Personal Characteristics- does not discuss race? Has this been shown to affect life satisfaction and loneliness? In general, I think the exclusion of this should be justified throughout the manuscript.

Thank you for your comment. Race is not included due to a lack of different ethnic backgrounds in the sample and is therefore deleted. A comment is added to the limitations section.

Materials and Methods

Lines 173-175- delete sentence outlining the rest of the section. This is unnecessary.

Thank you. The lines are deleted.

Conceptual Model- why did you not explore how personal characteristics influence mobility and neighborhood characteristics? We know that individuals who are low SES or lower educational status are more likely to live in disadvantaged neighborhoods, which may be characterized as having low social cohesion and low walkability.

Thank you for your question. We did not include these relationships since no disadvantaged neighbourhoods are selected for the study. Therefore, the relationships that may exist do not add to our existing knowledge of social cohesion and walkability in disadvantaged neighbourhoods.

Additionally, shouldn’t there a be a line from mobility to neighborhood characteristics? As you state in your introduction, an environment that is more walkable foster social interactions that in turn increase social cohesion. Therefore, you would hypothesize that mobility influences neighborhood characteristics.

Thank you for your suggestion. We chose to not include these relationships since the focus of current study is on how public space is used, instead of the characteristics of the neighbourhoods itself.

3.1.2- which of the four walkability scale items were included in the factor?

Thank you for your question. The four items are added to the section.

3.1.3.- These are all social environmental measures. I may suggest terming these “Social Neighbourhood Characteristics.” The physical environment is actually captured within your mobility construct as walkability. There is a little bit of disconnect between your literature review and study because you make a case for the importance of physical factors in the neighborhood environment under the Neighbourhood Characteristics section (2.3- lines 160-166) but that is not measured in this domain or could be argued that it is measured under mobility.

Thank you for pointing towards this disconnect. We used your suggestion for the term and deleted the lines that are about the physical characteristics of the neighbourhood.

3.1.4- is it not clear what was included after your principal component analysis from this text. The authors provide additional detail in the results section- however, these are methods. I would clarify briefly in the text here and add the details you provide in Lines 309-333 to an appendix.

Thank you for your suggestion. We added a more comprehensive explanation and made an appendix with the principal component analysis itself and the six components that arose.

3.2 Data Collection- You mention in later in the paper (section 4.1) that elderly populations were targeted, however that is not made clear in your data collection methods. Please clarify in this section whether elderly centers were targeted. These methods should appear here, then section 4.1 should present results only, the justification for why your study population is not representative of the Netherlands, etc should be in the discussion section.

Thank you for this comment. Elderly centres were used to obtain a higher number of elderly participants in the survey.

Also thanks for your suggestion to split the results and move the representativeness of the sample to the discussion section. We rewrote the results and discussion according to your suggestions.

Results

4.1- I love Table 1 and how it compares your population to that of the Netherlands and s-Hertogenbosch. These results should state the main differences between your study population and the other two population, but not discuss why this is the case. This discussion should be within your discussion section only.

Thank you! The explanation of the differences is moved to the discussion section.

Remove Lines 309-333 and make an appendix that is referenced within your methods section.

Thank you. We followed your suggestion.

Table 2- did you not ask about active transport (e.g biking or walking?) You will need to justify this.

Thank you. We did not include active transport modes, since they are already included in the three questions that were asked for public space use.

All results sections need to be majorly revised to simply state the results. The majority of your current results are actually discussion points and can be integrated into your current discussion.

Thanks for the suggestion. We made major revisions to the results and discussion according to your suggestions.

Discussion

Again, I would move literature support from your results sections to the discussion to highlight your key findings and add to your discussion. Otherwise, I like the discussion and conclusion. The summary of findings in the discussion is concise and helps consolidate findings for the reader.

Thank you for your comments. We split the results and discussion more clearly.

Should add consider adding the lack of examination of race to the limitations section. Additionally, how did a sample that was largely higher income and higher educational status influence findings?

Thank your for your suggestion. We added a section about the limitations of the sample.

Many thanks for your thorough review of our paper. We made major changes to the results and discussion section, according to your suggestions!